

# Awareness, behavior and attitudes concerning sun exposure among beachgoers in the northern coast of Peru

Carlos J. Toro-Huamanchumo[1], Sara J. Burgos-Muñoz[2], Luz M. Vargas-Tineo[2], Jhosuny Perez-Fernandez[2], Otto W. Vargas-Tineo[2], Ruth M. Burgos-Muñoz[2], Javier A. Zentner-Guevara[2] and Carlos Bada[3,4]

[1] Universidad de San Martín de Porres, Facultad de Medicina, Centro de Investigación en Epidemiología Clínica y Medicina Basada en Evidencias, Lima, Peru
[2] School of Medicine, Universidad de San Martín de Porres, Chiclayo, Peru
[3] School of Medicine, Universidad de San Martín de Porres, Lima, Peru
[4] Clínica San Judas Tadeo, Lima, Peru

Corresponding author
Carlos J. Toro-Huamanchumo,
carlos_toro@usmp.pe,
toro2993@hotmail.com

## ABSTRACT

**Background**. Skin cancer incidence has increased over the last years, becoming a major public health problem.
**Objective**. To describe the awareness, behavior and attitudes concerning sun exposure among beachgoers in the northern coast of Peru.
**Methods**. We conducted a cross-sectional study in the Pimentel beach, Peru. The "Beach Questionnaire" was used and we surveyed all the beachgoers from 8 a.m. to 4 p.m. and from March 5 to March 19. For the statistical analysis, sun exposure habits, sunburns history, knowledge, attitudes and practices were crossed with sex using the chi2 test.
**Results**. We surveyed 410 beachgoers, the most frequent phototype was type III (40.5%). Only the 13.66% of the respondents correctly answered the seven knowledge questions related to sun exposure and skin cancer. Men more frequently agreed that "when they are tanned their clothes looks nicer" ($p = 0.048$). Likewise, regarding the questions "Sunbathing is relaxing" and "Sunbathing improves my mood", men agreed or totally agreed with more frequency than women (63.64% vs. 46.15%, $p < 0.001$; and 61.36% vs 49.15%, $p = 0.014$, respectively). Regarding sun protection practices, women more frequently used sunshade ($p = 0.001$) and sunscreen (SPF ≥ 15) ($p < 0.001$) when compared to the male group.
**Conclusion**. Sun exposure is a potentially preventable risk factor for skin cancer. Thus, awareness of the risks of UVR overexposure and adequate sun-protective behaviors and attitudes are essential. Our results, however, are not as favorable as expected. Public health efforts should encourage sun-safety precautions and intervention campaigns should be carried out in recreational settings, such as the beaches.

## BACKGROUND

Skin cancer incidence has increased over the last years, becoming a major public health problem with a serious economic burden to the healthcare system of many countries (*Erdmann et al., 2013*; *Garbe & Leiter, 2009*; *Guy et al., 2015*). According to GLOBOCAN estimates, about 232.000 cases of melanoma and 55.000 deaths from this cause occurred worldwide in 2012 (*Ferlay et al., 2015*).

In recent years, global incidence rates of skin cancer have increased and there are some published reports that evidence this situation. For example, melanoma raw incidence rates per 100,000 US population has climbed from 22.2 to 23.6 (2009–2016 period). Similarly, raw mortality rates per 100 000 population has increased from 2.8 to 3.1 (*Glazer et al., 2017*). In Europe, melanoma trends has also increased in recent years, with the highest incidence rates in the UK, Ireland and the Netherlands (*Arnold et al., 2014*). Unfortunately, available data for Latin America is very limited (*Schmerling et al., 2011*). In Peru, there has been reported a growing trend of skin cancer, becoming the fourth most frequent type of cancer in the country (*Sordo & Gutiérrez, 2013*).

Sun exposure is considered a potentially preventable risk factor for skin cancer (*Molho-Pessach & Lotem, 2007*) and an adequate knowledge and good practices play an important role in the prevention of the disease. In fact, some studies have been carried out in order to assess these variables in patients, workers and students (*Thomas-Gavelan et al., 2011*; *Lucena et al., 2014*; *Hault et al., 2016*; *Gao, Liu & Liu, 2014*; *Fernández-Morano et al., 2017*; *Fernández-Morano et al., 2014*). However, only a few have focused on beachgoers, who are an important population at risk (*Cercato et al., 2015*; *Pagoto, McChargue & Fuqua, 2003*; *Weinstock et al., 2000*), and two of these studies only focused on behaviors and did not address knowledge or attitudes. In addition, the countries where these studies were conducted have a UV index lower than that reported in Peru (*Newman & McKenzie, 2011*).

In Peru, high temperature peaks have been reported over the last years, especially in 2017 (*Servicio Nacional de Meteorología e Hidrología, 2017*). In addition, Peru has been cataloged by the National Meteorology and Hydrology Service (SENAMHI) as one of the countries with the highest solar radiation, reaching an index of ultraviolet radiation (UV index) of 19 on a scale of 0 to 20 (*Servicio Nacional de Meteorología e Hidrología, 2017*). The northern coast of Peru has a semi-warm and tropical-dry climate where rainfall is barely present (*Oficina Nacional de Gobierno Electrónico e Informática, 2017*; *Feddema, 2005*). In summer, this region becomes even warmer, surpassing 30 °C (*Servicio Nacional de Meteorología e Hidrología, 2017*; *Oficina Nacional de Gobierno Electrónico e Informática, 2017*).

For the above mentioned, the objective of the present study was to describe the awareness, behavior and attitudes concerning sun exposure among beachgoers in the northern coast of Peru.

## METHODS

### Study design

We conducted a cross-sectional study in the Pimentel beach, Peru.

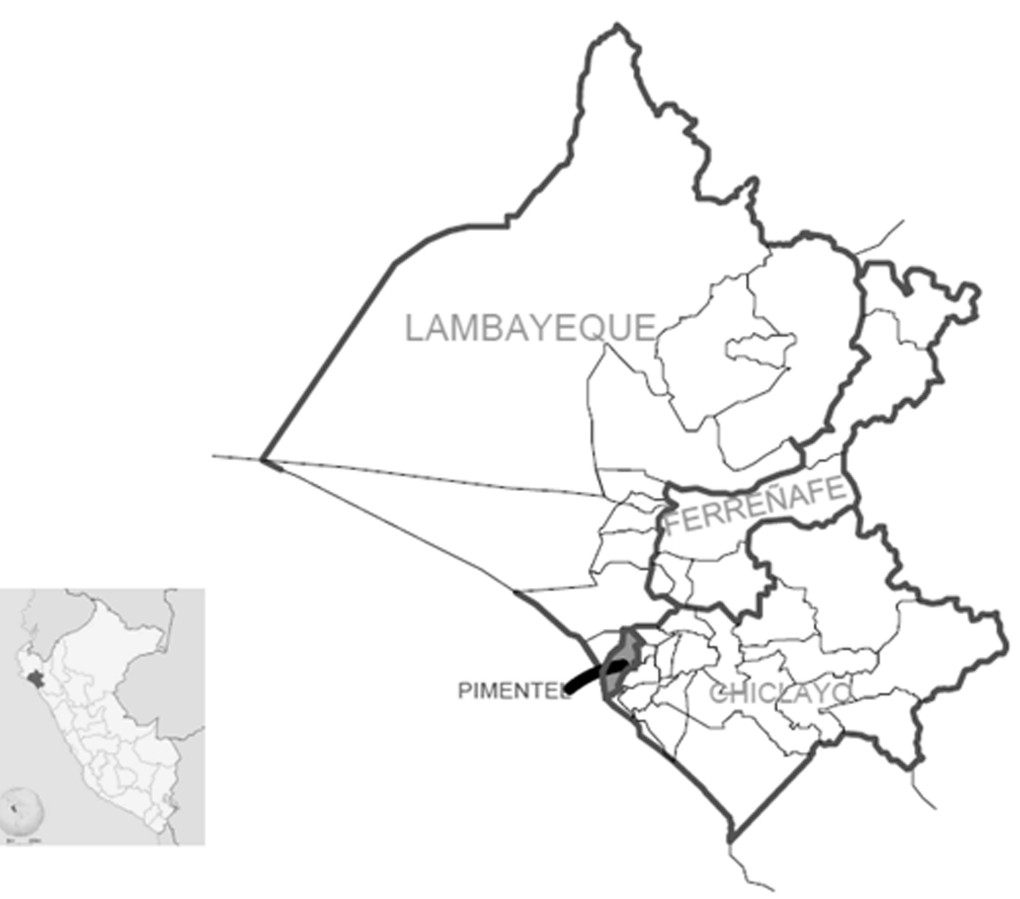

**Figure 1** **Map of the study area.**

## Setting and participants

Pimentel is one of the main beaches in the northern Peru and belongs to Lambayeque, which is considered a semi-warm and tropical-dry region, with temperatures that exceed 30 °C during summer (*Servicio Nacional de Meteorología e Hidrología, 2017*; *Oficina Nacional de Gobierno Electrónico e Informática, 2017*) (Fig. 1). We surveyed all the beachgoers from 8 a.m. to 4 p.m. and from March 5 to March 19 (Peruvian summer, 2018) (*Servicio Nacional de Meteorología e Hidrología, 2017*). No sample was calculated. We surveyed all Spanish-speaking adults aged 18–59 who were in the study place within the specified time range.

## Variables and data collection

We applied the "Beach Questionnaire", validated by *De Troya-Martín et al. (2009)* in a sample of Spanish beachgoers. This instrument aims to evaluate subjects' behavior, attitudes and knowledge regarding sun exposure, and has been used in previous studies with similar populations (*De Troya-Martín et al., 2014*; *De Troya-Martín et al., 2018*). It has also been shown to be valid, reliable (Cronbach $\alpha > 0.7$) and with good sensitivity to change (*De Troya-Martín et al., 2009*; *Fernández-Morano et al., 2015*).

The questionnaire included all our study variables and had the following sections: (1) Sociodemographic and academic data: sex, age, marital status, country of birth and educational level; (2) Color of non-sun-exposed skin: very fair, fair olive and dark; (3) Phototype: according to the Fitzpatrick model: I–IV, according to the erythema and tanning response after the first 60-min sun exposure in summer (*De Troya-Martín et al., 2018*); (4) Sun exposure habits on the beach in the last two summers: number of days spent at the beach each last two summers, number of hours per day and number of hours at midday (defined as between 12.00 and 16.00); (5) Sunburn history in the last summer (sunburn was defined as painful reddening of the skin) (*De Troya-Martín et al., 2018*); (6) Participants' general knowledge about sun exposure with dichotomous response (true or false); (7) Attitudes related with sun exposure and sun protection, on a Likert-like scale of five categories (from "totally disagree" to "totally agree") and (8) Sun protection practices.

## Statistical analysis

Collected data was entered into Microsoft Excel® with a double entry method to avoid errors during the process. After quality control, the database was exported to Stata version 13.0 (StataCorp LP, College Station, TX, USA).

We used relative and absolute frequencies to describe categorical variables and medians with interquartile ranges (after checking the absence of normality with Shapiro Wilk) for numerical variables. For bivariate analysis, we compared the categorical variables according to sex using the chi2 test. We considered a *P*-value < 0.05 as statistically significant.

## Ethics

This study was approved by the Institutional Review Board of the Hospital Nacional Docente Madre-Niño San Bartolome (RCEI-40), Lima, Peru. The participation was voluntary, and participants provided their informed oral consent, prior filling the survey. The anonymity of the participants and data confidentiality were ensured.

# RESULTS

## Baseline characteristics of the study population

We surveyed a total of 410 beachgoers. The most frequent skin colors were olive (46.6%) and pale (35.4%). The most frequent Fitzpatrick phototype was type III (40.5%). Detailed sociodemographic and academic data are shown in Table 1.

## Sun-exposure habits and sunburns history

Men went to the beach more frequently in the last two summers (20.46% went more than 15 days vs 12.82% of women, $p = 0.028$). Likewise, 62.2% of the participants reported having suffered at least one sunburn last summer (Table 2).

## Knowledge about sun exposure

Only the 13.66% of respondents ($n = 56$) correctly answered the seven questions related to sun exposure and skin cancer (Table 3). Individual analysis showed that the following questions had the lower percentage of correct answers: "Sun protection creams prevent aging of the skin produced by solar radiation" (60.0%) and "If I use total sun block I can

**Table 1  Sociodemographic, skin color and phototype data (n = 410).**

| Characteristics | n (%) |
| --- | --- |
| Sex | |
|     Male | 176 (42.9) |
|     Female | 234 (57.1) |
| Age (years)[a] | 28 (18–65) |
| Marital status | |
|     Single | 226 (55.1) |
|     Married or living w/partner | 175 (42.7) |
|     Separated/Divorced | 6 (1.5) |
|     Widowed | 3 (0.7) |
| Country of birth | |
|     Peru | 401 (98.1) |
|     Argentina | 2 (0.5) |
|     Colombia | 3 (0.7) |
|     Ecuador | 2 (0.5) |
|     Mexico | 1 (0.2) |
| Education | |
|     None | 4 (1.0) |
|     Primary | 11 (2.7) |
|     Secondary | 137 (33.4) |
|     Higher Education | 258 (62.9) |
| Skin color | |
|     Very fair | 22 (5.4) |
|     Fair | 145 (35.4) |
|     Olive | 191 (46.6) |
|     Dark | 52 (12.7) |
| Phototype | |
|     I | 62 (15.1) |
|     II | 79 (19.3) |
|     III | 166 (40.5) |
|     IV | 103 (25.1) |

**Notes.**
[a]Median (Interquartile range).

sunbathe without any risk'' (58.29%). Likewise, according to sex, significant differences were found in the response to "Once my skin is tanned, I don't need to use sun protection cream" (76.14% of men answered correctly, versus 88.46% of women, $p = 0.001$).

## Attitudes related with sun exposure

More than three quarters of the respondents agreed or totally agreed that it is necessary to use sunscreen creams to avoid problems in the future (90.49%) and that its use is worthwhile despite not getting a tan (77.80%) (Table 4). Men were more frequently agreed that when they are tanned their clothes looks nicer ($p = 0.048$). Likewise, regarding the question "Sunbathing is relaxing", men agreed or totally agreed with more frequency

**Table 2  Sun-exposure habits and sunburns history.**

| Item | Men n (%) | Women n (%) | Total n (%) | $p^a$ |
|---|---|---|---|---|
| *In relation with the last two summers, choose…* | | | | |
| Days of sun on the beach | | | | |
| None | 16 (9.09) | 40 (17.09) | 56 (13.66) | **0.028** |
| 1–5 | 95 (53.98) | 120 (51.28) | 215 (52.44) | |
| 6–15 | 29 (16.48) | 44 (18.80) | 73 (17.80) | |
| 16–30 | 17 (9.66) | 9 (3.85) | 26 (6.34) | |
| >30 | 19 (10.80) | 21 (8.97) | 40 (9.76) | |
| Hours of sun exposure on the beach | | | | |
| <30 min | 26 (14.77) | 41 (17.52) | 67 (16.34) | 0.706 |
| 30–1 h | 43 (24.43) | 52 (22.22) | 95 (23.17) | |
| 1–3 h | 65 (36.93) | 93 (79.74) | 158 (38.54) | |
| >3 h | 42 (23.86) | 48 (20.51) | 90 (21.95) | |
| Hours of sun at midday | | | | |
| No sun | 24 (13.64) | 42 (17.95) | 66 (16.10) | 0.184 |
| <1 h | 48 (27.27) | 42 (17.95) | 90 (21.95) | |
| 1–2 h | 38 (21.59) | 61 (26.07) | 99 (24.15) | |
| 2–4 h | 39 (22.16) | 56 (23.93) | 95 (23.17) | |
| 4–6 h | 27 (15.34) | 33 (14.10) | 60 (14.63) | |
| *Last summer…* | | | | |
| Sunburns | | | | |
| None | 63 (35.80) | 92 (39.32) | 155 (37.80) | 0.818 |
| 1–2 | 73 (41.48) | 100 (42.74) | 173 (42.20) | |
| 3–5 | 28 (19.51) | 29 (12.39) | 57 (13.90) | |
| 6–10 | 5 (2.84) | 5 (2.14) | 10 (2.44) | |
| >10 | 7 (3.98) | 8 (3.42) | 15 (3.66) | |

**Notes.**
[a] Chi2 test.

than women (63.64% vs. 46.15%, $p < 0.001$). The same thing happened with the item "Sunbathing improves my mood" (61.36% of men vs 49.15% of women, $p = 0.014$).

## Sun protection practices

The 63.9% of the respondents indicated that they usually or always use sunscreen when they go to the beach (Table 5). However, the compliance percentage was lower for the rest of the practices. Analysis by sex showed that women more frequently used sunshade ($p = 0.001$) and sunscreen (SPF ≥ 15) ($p < 0.001$).

## DISCUSSION

### Sun-exposure habits and sunburns history

In our study, we found that men went with more frequency to the beach than women, which may be related to recreational activities that are often performed at the place of study (e.g., surfing and soccer). This finding differs from that found by *Fernández-Morano et al. (2014)*. In their study, women went to the beach more frequently (75.5% compared

**Table 3  Participants' general knowledge about sun exposure.**

| Item | Men n (%) | Women n (%) | Total n (%) | p[a] |
|---|---|---|---|---|
| Sun protection creams prevent aging of the skin produced by solar radiation | | | | |
| True | 107 (60.80) | 139 (59.40) | 246 (60.0) | 0.776 |
| False | 69 (39.20) | 95 (40.60) | 164 (40.0) | |
| Sun is the main cause of skin cancer | | | | 0.452 |
| True | 163 (92.61) | 221 (94.44) | 384 (93.66) | |
| False | 13 (7.39) | 13 (5.56) | 26 (6.34) | |
| Sun produces marks on the skin | | | | 0.135 |
| True | 152 (86.36) | 213 (91.03) | 365 (89.02) | |
| False | 24 (13.64) | 21 (8.97) | 45 (10.98) | |
| If I use sunscreen I can sunbathe without any risk | | | | |
| True | 79 (44.89) | 92 (39.32) | 171 (41.71) | 0.258 |
| False | 97 (55.11) | 142 (60.68) | 239 (58.29) | |
| Avoiding the midday sun (11–17 h) is the most efficient way of protecting my skin | | | | |
| True | 137 (77.84) | 176 (75.21) | 313 (76.34) | 0.536 |
| False | 39 (22.16) | 58 (24.79) | 97 (23.66) | |
| Once my skin is tanned, I don't need to use sun protection cream | | | | |
| True | 42 (23.86) | 27 (11.54) | 69 (16.83) | **0.001** |
| False | 134 (76.14) | 207 (88.46) | 341 (83.17) | |

**Notes.**
[a]Chi2 test.

to 66.4% of men) (*Fernández-Morano et al., 2014*). However, this may be because its population was comprised only of adolescents, which may be related to another of their findings, which was a higher likelihood for sunbathing and tanning by the female group.

We found that more than 60% had suffered at least one sunburn in the last summer, a percentage higher than those reported in studies conducted in the US (*Buller et al., 2011*; *Holman et al., 2014*) and Europe (*De Troya-Martín et al., 2018*; *Haluza et al., 2016*; *Kritsotakis et al., 2016*). This may be due to the lack of education in the local population, which negatively affects their practices and habits regarding sun exposure. This finding is a call for the implementation of intervention and education strategies, since it has been demonstrated that a personal sunburns history is strongly associated with skin cancer (*Erdmann et al., 2013*; *Garbe & Leiter, 2009*; *Sánchez, Nova & De la Hoz, 2012*; *Wu et al., 2016*).

## Participants' general knowledge about sun exposure

Less than 15% of the respondents correctly answered the seven questions about sun exposure. This lack of knowledge could be a possible explanation for the growing trend of skin cancer in the Peruvian population (*Sordo & Gutiérrez, 2013*). Studies conducted in adolescents and adults beachgoers in Spain have reported better levels of knowledge (*Fernández-Morano et al., 2014*; *Cercato et al., 2015*), which could be a reflection

**Table 4  Attitudes related with sun exposure.**

| Item | Men n (%) | Women n (%) | Total n (%) | p[a] |
|---|---|---|---|---|
| When I am tanned my clothes look nicer | | | | |
| Totally agree/Agree | 61 (34.66) | 60 (25.64) | 121 (29.51) | **0.048** |
| Indifferent/Disagree/Totally disagree | 115 (65.34) | 174 (74.36) | 289 (70.49) | |
| Sunbathing helps prevent health problems | | | | |
| Totally agree/Agree | 76 (43.18) | 109 (46.58) | 185 (45.12) | 0.494 |
| Indifferent/Disagree/Totally disagree | 100 (56.82) | 125 (53.42) | 225 (54.88) | |
| I like the feeling of the sun on my skin when I am lying on the beach | | | | |
| Totally agree/Agree | 64 (36.36) | 65 (27.78) | 129 (31.46) | 0.064 |
| Indifferent/Disagree/Totally disagree | 112 (63.64) | 169 (72.22) | 281 (54.88) | |
| It is worth using sun protection cream to avoid future problems | | | | |
| Totally agree/Agree | 161 (91.48) | 210 (89.74) | 371 (90.49) | 0.554 |
| Indifferent/Disagree/Totally disagree | 15 (8.52) | 24 (10.26) | 39 (9.51) | |
| I find sun protection creams unpleasant | | | | |
| Totally agree/Agree | 52 (29.55) | 69 (29.49) | 121 (29.51) | 0.990 |
| Indifferent/Disagree/Totally disagree | 124 (70.45) | 165 (70.51) | 289 (70.49) | |
| It is worth using sun protection cream even though I don't get a tan | | | | |
| Totally agree/Agree | 136 (77.27) | 183 (78.21) | 319 (77.80) | 0.822 |
| Indifferent/Disagree/Totally disagree | 40 (22.73) | 51 (21.79) | 91 (22.20) | |
| People with a tan are more attractive | | | | |
| Totally agree/Agree | 67 (38.07) | 79 (33.76) | 146 (35.61) | 0.367 |
| Indifferent/Disagree/Totally disagree | 109 (61.93) | 155 (66.24) | 264 (64.39) | |
| Sunbathing is healthy for my body | | | | |
| Totally agree/Agree | 95 (53.98) | 107 (45.73) | 202 (49.27) | 0.098 |
| Indifferent/Disagree/Totally disagree | 81 (46.02) | 127 (54.27) | 208 (50.73) | |
| Sunbathing is relaxing | | | | |
| Totally agree/Agree | 112 (63.64) | 108 (46.15) | 220 (53.66) | **<0.001** |
| Indifferent/Disagree/Totally disagree | 64 (36.36) | 126 (53.85) | 190 (46.34) | |
| Having a tan makes you look young and relaxed | | | | |
| Totally agree/Agree | 63 (35.80) | 67 (28.63) | 130 (31.71) | 0.123 |
| Indifferent/Disagree/Totally disagree | 113 (64.20) | 167 (71.37) | 280 (68.29) | |
| Sunbathing improves my mood | | | | |
| Totally agree/Agree | 108 (61.36) | 115 (49.15) | 223 (54.39) | **0.014** |
| Indifferent/Disagree/Totally disagree | 68 (38.64) | 119 (50.85) | 187 (45.61) | |
| I like sunbathing | | | | |
| Totally agree/Agree | 108 (61.36) | 121 (51.71) | 229 (55.85) | 0.051 |
| Indifferent/Disagree/Totally disagree | 68 (38.64) | 113 (48.29) | 181 (44.15) | |
| When I go to the beach I prefer to be in the shade | | | | |
| Totally agree/Agree | 128 (72.73) | 173 (73.93) | 301 (73.41) | 0.785 |

*(continued on next page)*

**Table 4** (*continued*)

| Item | Men<br>*n* (%) | Women<br>*n* (%) | Total<br>*n* (%) | *p*[a] |
|---|---|---|---|---|
| Indifferent/Disagree/Totally disagree | 48 (27.27) | 61 (26.07) | 109 (26.59) | |
| I don't like high-protection creams because they are anti-aesthetic | | | | |
| Totally agree/Agree | 53 (30.11) | 62 (26.50) | 115 (28.05) | 0.420 |
| Indifferent/Disagree/Totally disagree | 123 (69.89) | 172 (73.50) | 295 (71.95) | |

**Notes.**
[a]Chi2 test.

of the positive impact of the interventions and campaigns that have been carried out in that country (*De Troya-Martín et al., 2014*; *Fernández-Morano et al., 2015*; *Del Boz et al., 2015*).

Some studies suggest that a good level of knowledge about sun exposure may not always go hand in hand with adequate attitudes or practices (*Haluza, Simic & Moshammer, 2016*; *Mousavi et al., 2011*; *Yan et al., 2015*). However, a systematic review showed that some sun protection behaviors were positively associated with a good level of knowledge about skin cancer (*Day et al., 2014*). Also, a study conducted by *De Troya-Martín et al. (2018)* reported an important role of knowledge about sun exposure in the prevention of sunburns.

## Attitudes related with sun exposure

Most of the participants presented good attitudes regarding the use of sunscreen. These results are more favorable to those found by *Mousavi et al. (2011)* and *Fernández-Morano et al. (2017)*. A possible explanation may be that during the last summer, high temperature peaks were reported in comparison to previous years, as well as heavy rains on the northern coast of Peru (*Servicio Nacional de Meteorología e Hidrología, 2017*). To face the problem, intervention, reconstruction and prevention activities were carried out, including information campaigns which were disseminated by local and national media.

On the other hand, men presented inappropriate attitudes more frequently, which differs from the results reported in two studies conducted in Spain (*Fernández-Morano et al., 2017*; *Fernández-Morano et al., 2014*). A possible explanation may lie in the continuing influence of social media on current stereotypes and the perception of beauty and body image concerns (*Fardouly & Vartanian, 2016*; *Barlett, Vowels & Saucier, 2008*). In this sense, since tanning has usually been related to concepts of beauty, our result could be understood a little more by the fact that nowadays men are increasingly adopting some attitudes that were previously prioritized by women, such as sunbathing and tanning (*Fernández-Morano et al., 2017*; *Haluza et al., 2016*).

## Sun protection practices

More than half of the participants reported a frequent sunscreen use (usually or always). This result is similar to that found by *Devos et al. (2012)* in a sample of beachgoers from the northern coast of Belgium, and better than those reported in other studies conducted in Europe (*Fernández-Morano et al., 2014*; *Weinstock et al., 2000*) and Asia (*Mousavi et al., 2011*; *Yan et al., 2015*). However, the percentages of compliance for the other practices were less than 50%. Since current literature mentions that sunscreen use alone

**Table 5   Sun protection practices.**

| Item | Men n (%) | Women n (%) | Total n (%) | $p^a$ |
|---|---|---|---|---|
| *When you go to the beach, you…* | | | | |
| Use sunshade | | | | |
| Always | 37 (21.02) | 92 (39.32) | 129 (31.46) | **0.001** |
| Usually | 27 (15.34) | 27 (11.54) | 54 (13.17) | |
| Sometimes | 50 (28.41) | 63 (26.92) | 113 (27.56) | |
| Almost never | 25 (14.20) | 25 (10.68) | 50 (12.20) | |
| Never | 37 (21.02) | 27 (11.54) | 64 (15.61) | |
| Use sunglasses | | | | |
| Always | 35 (19.89) | 65 (27.78) | 100 (24.39) | 0.406 |
| Usually | 23 (13.07) | 28 (11.97) | 51 (12.44) | |
| Sometimes | 43 (24.43) | 58 (24.79) | 101 (24.63) | |
| Almost never | 25 (14.20) | 28 (11.97) | 53 (12.93) | |
| Never | 50 (28.41) | 55 (23.50) | 105 (25.61) | |
| Use hat or cap | | | | |
| Always | 57 (32.39) | 73 (31.20) | 130 (31.71) | 0.718 |
| Usually | 28 (15.91) | 35 (14.96) | 63 (15.37) | |
| Sometimes | 37 (21.02) | 60 (25.64) | 97 (23.66) | |
| Almost never | 20 (11.36) | 30 (12.82) | 50 (12.20) | |
| Never | 34 (19.32) | 36 (15.38) | 70 (17.07) | |
| Wear long sleeves or long trousers | | | | |
| Always | 22 (12.50) | 27 (11.54) | 49 (11.95) | 0.742 |
| Usually | 17 (9.66) | 16 (6.84) | 33 (8.05) | |
| Sometimes | 29 (16.48) | 48 (20.51) | 77 (18.78) | |
| Almost never | 33 (18.75) | 45 (19.23) | 78 (19.02) | |
| Never | 75 (42.61) | 98 (41.88) | 173 (42.20) | |
| Avoid sun 12.00–16.00 | | | | |
| Always | 38 (21.59) | 60 (25.64) | 98 (23.90) | 0.802 |
| Usually | 32 (18.18) | 44 (18.80) | 76 (18.54) | |
| Sometimes | 61 (34.66) | 72 (30.77) | 133 (32.44) | |
| Almost never | 15 (8.52) | 23 (9.83) | 38 (9.27) | |
| Never | 30 (17.05) | 35 (14.96) | 65 (15.85) | |
| Use sunscreen (SPF ≥ 15) | | | | |
| Always | 54 (30.68) | 125 (53.42) | 179 (43.66) | **<0.001** |
| Usually | 37 (21.02) | 46 (19.66) | 83 (20.24) | |
| Sometimes | 42 (23.86) | 43 (18.38) | 85 (20.73) | |
| Almost never | 14 (7.95) | 10 (4.27) | 24 (5.85) | |
| Never | 29 (16.48) | 10 (4.27) | 39 (9.51) | |

**Notes.**
SPF,  Sun Protection Factor.
[a] Chi2 test.

is not enough to control the skin exposure to UVR (*Diffey, 2001*; *Iannacone, Hughes & Green, 2014*; *Grossman et al., 2018*), beachgoers should adopt other measures, such as avoiding midday, wearing hat/cap and long-sleeved clothes, seeking for shade and skin

self-examination (*Molho-Pessach & Lotem, 2007*; *Grossman et al., 2018*; *Skotarczak et al., 2015*; *Mancebo, Hu & Wang, 2014*).

Women had more and better sun protection practices, mainly related to the use of sunscreen and sunshade. Previous research have also reported better sun exposure behaviors and practices in this population (*Weinstock et al., 2000*; *Yan et al., 2015*; *Devos et al., 2012*; *Olsen et al., 2015*). This may be linked to the attitudes that, according to our study, were also better in women. In addition, this could explain why sunburns are more frequent in men, according to some studies (*De Troya-Martín et al., 2018*; *Reuter et al., 2010*; *Kasparian, McLoone & Meiser, 2009*).

### Relevance and implications

Skin cancer has become a major public health problem (*Erdmann et al., 2013*; *Garbe & Leiter, 2009*; *Guy et al., 2015*). In recent years, its global incidence rates have increased (*Glazer et al., 2017*; *Arnold et al., 2014*; *Schmerling et al., 2011*), and Peru is not the exception (*Sordo & Gutiérrez, 2013*). Thus, awareness of the risks of sun exposure and adequate sun-protective behaviors and attitudes are needed. Our results, however, are not as favorable as expected.

Evidence suggest that Public Health efforts should encourage sun-safety precautions to avoid UVR overexposure (*Haluza et al., 2016*; *Blumthaler, 2018*). In addition, the beach seems to be an ideal setting for promoting adequate sun-protective behaviors (*Cercato et al., 2015*). In this sense, prevention, detection and intervention campaigns related to sun protection and skin cancer should be carried out, as they have shown satisfactory results in other studies (*Pagoto, McChargue & Fuqua, 2003*; *De Troya-Martín et al., 2014*; *Del Boz et al., 2015*; *Emmons et al., 2011*).

### Limitations

Some limitations must be highlighted. First, we used self-report questions; despite using a validated instrument to measure our variables, social desirability bias might arise. Second, we did not address some variables that could potentially influence the results of our study, such as current or previous illnesses and family history of skin cancer. Finally, the extrapolation of our results is limited to the Pimentel beachgoers. However, given that it is the busiest beach in Lambayeque, it gives us a good approximation to the possible reality in the region.

## CONCLUSION

Only one or two out of ten respondents correctly answered all the questions related to sun exposure knowledge. Negative attitudes were more frequent in men, and women presented better practices. Future research should study other variables that are also related to sun protection. Thus, interventions could be more targeted and with even more promising results. Finally, we recommend that future studies develop and evaluate the impact of sun-protective interventions, as previous research have shown their potential to promote sun protection in recreational settings (*Hay et al., 2017*; *Rodrigues et al., 2017*; *Rodrigues, Sniehotta & Araujo-Soares, 2013*).

### Funding

The authors received no funding for this work.

### Competing Interests

The authors declare there are no competing interests.

### Author Contributions

- Carlos J. Toro-Huamanchumo conceived and designed the experiments, analyzed the data, contributed reagents/materials/analysis tools, prepared figures and/or tables, authored or reviewed drafts of the paper, approved the final draft.
- Sara J. Burgos-Muñoz conceived and designed the experiments, performed the experiments, prepared figures and/or tables, authored or reviewed drafts of the paper, approved the final draft.
- Luz M. Vargas-Tineo, Jhosuny Perez-Fernandez, Otto W. Vargas-Tineo, Ruth M. Burgos-Muñoz and Javier A. Zentner-Guevara performed the experiments, authored or reviewed drafts of the paper, approved the final draft.
- Carlos Bada analyzed the data, contributed reagents/materials/analysis tools, authored or reviewed drafts of the paper, approved the final draft.

### Human Ethics

The following information was supplied relating to ethical approvals (i.e., approving body and any reference numbers):

The Institutional Review Board of the Hospital Nacional Docente Madre-Niño San Bartolome (RCEI-40) granted Ethical approval to carry out the study (Ethical Application Ref: Exp.No 02770-17).

### Data Availability

Toro-Huamanchumo, Carlos (2018): Dataset.eng - Sun exposure. figshare. Dataset. https://doi.org/10.6084/m9.figshare.6828725.v1.

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
