# Peer review of "Awareness, behavior and attitudes concerning sun exposure among beachgoers in the northern coast of Peru"

_PeerJ, doi:10.7717/peerj.6189_

## Round 0.1 · original submission · Major Revisions

Your manuscript has been reviewed and several issues of concern have been raised. Please provide a point by point response regarding how each issue was addressed in the revised manuscript.

·

Basic reporting

Please see comments below.

Experimental design

Please see comments below.

Validity of the findings

Please see comments below.

Additional comments

General comments
=============
This manuscript aims to describe the awareness, behavior and attitudes concerning sun exposure among beachgoers in the northern coast of Peru. I would like to congratulate the authors for their efforts in planning the research and writing this manuscript.


Specific comments
=============

Major comments
* * *
1. Additional detail about the methods used would benefit his paper. My understanding is that there a checklist for reporting cross sectional studies (STROBE Statement), including all the recommended details on the STROBE statement would be useful.
2. I appreciate the restrictions in terms of length of the manuscript. However, I suggest that you improve your introduction by detailing the following:
2.1 In line 53, there is a mention to skin cancer trend. I would recommend highlighting in that first sentence what is the trend and then going to specific details by country.
2.2 A brief summary of previous studies would benefit the description at lines 58- 61 to provide more justification for your study (specifically, you should expand upon the knowledge gap being filled)
3. Given the goal of changing sun-safety behaviour in recreational settings, I suggest providing more background for this in your introduction. Rodrigues et al (Annals of Behavioral Medicine, 2012, V45, p224; JMIR research protocols, 2017, V6)) have shown the potential of behavioural interventions to promote sun protection in recreational settings. Please consider adding these details to your introduction section.
4. The results section needs more detail. I suggest that you improve the description at lines 124- 128 to provide more detail of what these figures mean in terms of the knowledge about sun exposure.
5. The discussion section seems to be missing information on what are the implications of your findings. Additional detail would help put this study in perspective for future interventions/prevention strategies.

Minor comments
* * *
6. The English language should be improved to ensure that an international audience can clearly understand your text. Some examples where the language could be improved include lines 38 and 198– the current phrasing makes comprehension difficult.
7. The abstract could be improved by revising the conclusions section. The first sentence is still related to results and a statement about implications of this study might be a better description.
8. The citations in text seem to be incorrect at times (line 170, 175). I would suggest checking these and your list of reference thoroughly.

Reviewer 2 ·

Basic reporting

No comment

Experimental design

No comment

Validity of the findings

No comment

Additional comments

The present study has investigated sun behavior among beachgoers. An important topic as skin cancer is a big problem worldwide. The manuscript is well-structured. The survey is based on a questionnaire.

My biggest concern with this study is the quality of this questionnaire especially the part about “knowledge about sun exposure”. These questions should deal with basic knowledge about sun exposure and skin cancer. However, I find it very difficult to answer these questions and the correct answers are not provided in the article. I think the weaknesses of these questions should be discussed in the discussion. E.g. I find the following questions difficult to answer:

- “If I use total sun block I can sunbath without any risk”
To use “total sun block” is a very speculative situation. There is no sunscreen that blocks all radiation and no sunscreen users apply sunscreen to their all their exposed skin without missed area. The word “total sun block” is not a good word to use.

- “Avoiding the sun at young ages (under 18 years) reduces the risk of skin cancer by 80%”
What is the right answer to this question? How do you know?

I would not recommend use of this part of the questionnaire again.

I have a few additional comments:
- What is meant by ”negative attitudes” in the following sentence in the abstract? “Negative attitudes were more frequent in men and women presented better”.
- Figure 1 could be omitted.
- In Table 2 it is not clear what questions the participants had answered.
- What year are your data collected?
-
Thanks for the opportunity to read this interesting manuscript.

---

## Round 0.2 · accepted · Accept

The initial reviewers have not provided their feedback on the revised version; however, my evaluation of the rebuttal letter and revisions to the manuscript indicate that the reviewers concerns have been addressed. I have consulted with the journal editorial board and I am happy to approve the manuscript.

#